# Move-Then-Operate: Behavioral Phasing for Human-Like Robotic Manipulation

Haoming Xu [1 2]   Lei Lei [3 4 2]   Jie Gu [2]   Chu Tang [2]   Jingmin Chen [2]   Rui-Qi Wang [5]

## Abstract

We present Move-Then-Operate, a Vision language action framework that explicitly decouples robotic manipulation into two distinct behavioral phases: coarse relocation (move) and contact-critical interaction (operate). Unlike monolithic policies that conflate these heterogeneous regimes, our architecture employs a dual-expert policy routed by a learnable phase selector, introducing a structural inductive bias that isolates phase-specific dynamics. Phase labels are automatically generated via an MLLM-based pipeline conditioned on lightweight contextual cues such as end-effector velocity and subtask decomposition to ensure alignment with human motor patterns. Evaluated on the RoboTwin2 benchmark, our method achieves an average success rate of 68.9%, outperforming the monolithic $\pi_0$ baseline by 24%. It matches or exceeds models trained on $10\times$ more data and reaches peak performance in 40% fewer training steps, demonstrating that architectural disentanglement of move and operate phases is a highly effective and efficient strategy for mastering high-precision manipulation.

## 1. Introduction

Vision–language–action (VLA) models have achieved strong performance across diverse robotic manipulation tasks (Black et al., 2024; Kim et al., 2024). Most contemporary systems learn monolithic policies that generate full action trajectories (Li et al., 2025d;b; Shukor et al., 2025b; Huang et al., 2025b), without distinguishing coarse

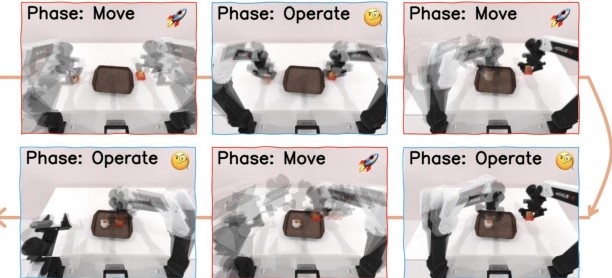

*Figure 1.* Schematic of robotic manipulation phases via equidistant sampling. The sequence comprises a fast and rapid move phase characterized by large displacements, and an operate phase focused on fine-grained and tenuous precision.

relocation from fine manipulation. Even dual-system variants that decompose tasks into sub-tasks still optimize all low-level actions under a unified objective (Bjorck et al., 2025). Long-range motion and contact-rich dexterity are thus tightly coupled within such a formulation.

Human manipulation, however, is widely characterized by a two-phase organization, as illustrated in Fig.2(a): an initial move phase that rapidly brings the effector into the vicinity of the target, followed by a feedback-driven operate phase that performs precise, contact-critical adjustments (Woodworth, 1899; Elliott et al., 2010). This division is also supported by classic speed–accuracy accounts such as Fitts' law, where coarse movement amplitude and final target tolerance differentially influence movement time (Fitts, 1954; Meyer et al., 1988). Notably, this two-phase structure also manifests in modern robotic datasets: as shown in Fig.2(c), actions from two phases exhibit distinct characteristics.

Recent studies have begun addressing trajectory heterogeneity through various strategies: SP-VLA (Li et al., 2025e) and Action-Aware Pruning (Pei et al., 2025b) focus on controller selection and compression. STARE-VLA (Xu et al., 2025) and Mixture of Horizons (Jing et al., 2025) explore stage-wise reinforcement and multi-horizon mixing to improve credit assignment. FedVLA (Miao et al., 2025) proposes a dual-stage MoE architecture to effectively learn across diverse tasks. Similarly, AdaMoE (Shen et al., 2025), MoE-DP (Cheng et al., 2025), and VER (Wang et al., 2025b) utilize action- or vision-specialized MoEs to better handle task heterogeneity. Despite these advances, prior efforts

Work completed during the internship at Rightly Robotics. [1]Hangzhou Institute for Advanced Study, University of Chinese Academy of Sciences, Hangzhou, China. [2]Rightly Robotics, Hangzhou, China. [3]Shanghai Innovation Institute, Shanghai, China. [4]University of Science and Technology of China, Hefei, China. [5]School of Aritificial intelligence, Institute of Artificial Intelligence, University of Science and Technology Beijing, Beijing, China. Correspondence to: Jie Gu <jgu@rightly.ai>.

*Proceedings of the 43rd International Conference on Machine Learning*, Seoul, South Korea. PMLR 306, 2026. Copyright 2026 by the author(s).

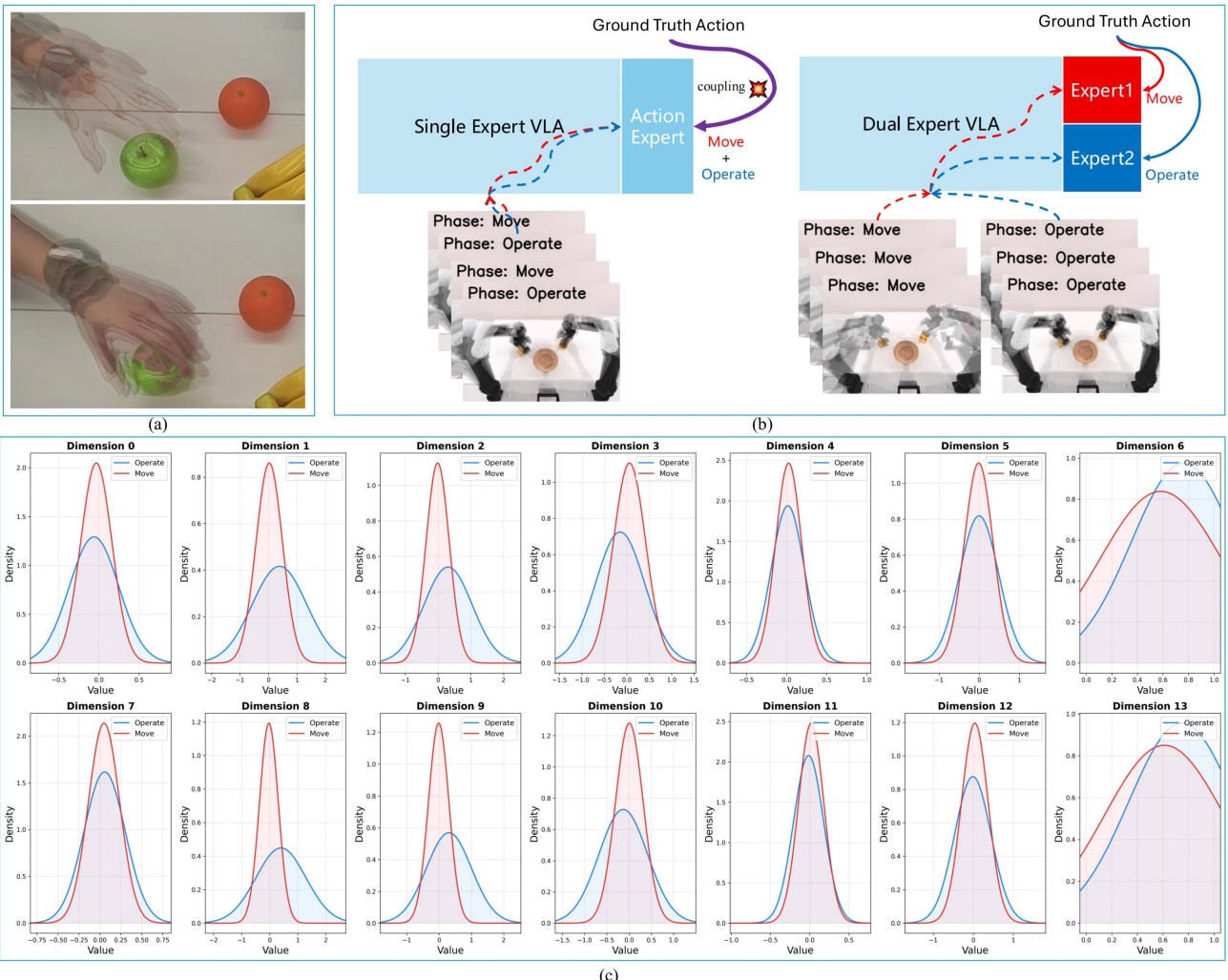

*Figure 2.* (a) Schematic representation of a two-phase process superimposed at uniform intervals and frequencies from human manipulation videos. The upper panel depicts a rapid approach toward the object, while the lower panel illustrates slow and meticulous adjustments. (b) A schematic diagram of our motivation. The amplitudes vary across different phases; specifically, joint learning and normalization cause actions in the operate phase to be overshadowed by the larger-magnitude movements of the move phase. Our approach of decoupled learning effectively addresses this issue. (c) Visualization of RoboTwin2 (Chen et al., 2025) training data annotations across two manipulation types, following the protocol in Section 4.4. Consistent with the z-score normalization employed in flow matching, we model the 14-DoF dual-arm actions as Gaussian distributions; note that the gripper states (the final dimension for each arm) do not strictly adhere to this assumption. Value is the absolute value of action. From figure, the Move phase (red) displays a broader amplitude and a more concentrated distribution than the Operate phase (blue).

primarily target RL signals or temporal scaling, and do not explicitly disentangle long-range move from contact-rich operate under predominantly imitation-based supervision.

To address this challenge, we introduce Move-Then-Operate, a Vision–Language–Action (VLA) framework that mirrors human motor strategies by explicitly decomposing tasks into distinct *move* and *operate* phases (Fig.2b). Our architecture employs a dual-expert policy, selected by a phase selection router, introducing a structural inductive bias that decouples coarse relocation from fine-grained manipulation. This design effectively mitigates the optimization instability and

gradient interference arising from the overwhelming prevalence of *move* frames (which may drown out the learning signal for manipulation), thereby facilitating the learning of their distinct action distributions. Consequently, this isolation not only enables clean behavioral disentanglement but also significantly improves both data and training efficiency.

To enable phase-aware training in Move-Then-Operate, we establish an automated pipeline for annotating move and operate phases in demonstration trajectories. A Multimodal Large Language Model (MLLM) is leveraged to perform segmentation. To align the segmentation with human oper-

ational patterns, we incorporate contextual cues (*e.g.*, subtask decomposition, phase transition heuristics, and velocity cues) as explicit constraints in the MLLM prompt.

In our experiments, evaluations on eight tasks in RoboTwin2 (Chen et al., 2025) benchmark demonstrate that our method significantly outperforms the monolithic $\pi_0$ baseline, achieving an average success rate of $68.9\%$ with a $+24.1\%$ improvement. Notably, our model exhibits superior data efficiency, rivaling or even surpassing baselines trained on $10\times$ more demonstrations. Furthermore, the proposed decoupled architecture demonstrates remarkable training efficiency, reaching peak performance in $40\%$ fewer iterations compared to the standard training schedule (full training budget). These results underscore that architectural decoupling of the move and operate phases is a highly effective strategy for mastering complex, high-precision robotic skills.

In summary, our primary contributions are as follows:

1. We propose **Move-Then-Operate**, a human-inspired VLA framework that explicitly disentangles long-range relocation from contact-rich manipulation through a dual-expert policy, effectively mitigating optimization interference and enhancing high-precision control.

2. We develop an automated, MLLM-based data annotation pipeline that incorporates subtask decomposition and contextual cues (*e.g.*, velocity cues) to generate high-fidelity phase labels that align with human operational patterns.

3. We demonstrate through extensive experiments on RoboTwin2 that our method achieves an $24.1\%$ absolute improvement in success rate over monolithic baselines, while exhibiting superior data and training efficiency—rivaling models trained on $10\times$ more demonstrations and reaching peak performance in $40\%$ fewer training steps.

## 2. Related Work

### 2.1. Vision-Language-Action Models

Vision-Language-Action (VLA) models have transformed robotic manipulation by unifying perception, semantic reasoning, and control into an end-to-end framework (Brohan et al., 2023; Zitkovich et al., 2023; Driess et al., 2023; Ma et al., 2024). While foundational autoregressive architectures leveraged web-scale knowledge by discretizing continuous actions into language-aligned tokens (Kim et al., 2024; Qu et al., 2025; Hung et al., 2025), recent advancements increasingly adopt diffusion and flow-matching policies to address the precision loss inherent in quantization (Ghosh et al., 2024; Black et al., 2024; Liu et al., 2025b; Chi et al., 2023). Hybrid approaches further attempt to bridge these

paradigms by combining the semantic planning strengths of transformers with the dense control signals of generative models (Liu et al., 2025a; Li et al., 2025a; Wen et al., 2025), yet they often face trade-offs between inference latency and trajectory smoothness in contact-rich manipulation tasks (Shen et al., 2025).

To mitigate the prohibitive computational demands of large multimodal backbones, a significant body of research has focused on architectural efficiency and inference acceleration (Pertsch et al., 2025; Li et al., 2025e; Huang et al., 2026). Dynamic computation strategies, such as adaptive token pruning and budget-aware visual processing, allow models to discard redundant information during static manipulation phases (Li et al., 2025e; Pei et al., 2025a), whereas Mixture-of-Experts (MoE) architectures scale parameter capacity efficiently by activating only task-relevant modules during forward passes (Shen et al., 2025). Parallel efforts investigating policy distillation and asynchronous execution frameworks demonstrate that optimized lightweight models can rival billion-parameter baselines in real-time control loops through aggressive quantization or architectural simplification (Shukor et al., 2025a; Wang et al., 2025a; Lin et al., 2025).

Beyond computational metrics, recent works tackle the temporal complexity of long-horizon tasks through hierarchical and spatio-temporal modeling (Jing et al., 2025; Xu et al., 2025; Ichter et al., 2022). Approaches leveraging horizon mixing or hierarchical decomposition explicitly separate high-level reasoning from low-level execution to improve credit assignment and generalization across diverse embodiments (Li et al., 2025c; Huang et al., 2025a; 2023). Despite these advances in temporal abstraction, the specific challenge of explicitly disentangling coarse-grained transport motions from precision-demanding interaction phases remains under-explored, as the dominance of simple reaching behaviors in human demonstrations often obscures the learning of fine-grained manipulation dynamics.

### 2.2. Dual-Expert and Decoupled Architectures

It is a common practice to decompose tasks in order to avoid conflicting objectives. For example, DeCoOP (Zhou et al., 2024) points out that using a single expert to solve out-of-distribution problems can also lead to conflicts. Long tailed recognition mirrors the imbalance between abundant move segments and sparse operate segments in robot demonstrations, and BBN (Zhou et al., 2020) shows that decoupling can separate broad feature learning from tail focused decision refinement, using a cumulative schedule that shifts from head dominated learning to tail oriented correction. Recent work replaces static splits with dynamic routing, DQRoute (Wei et al., 2025) allocates samples to experts by online difficulty and uncertainty, and ExPaMoE (Zhao et al.,

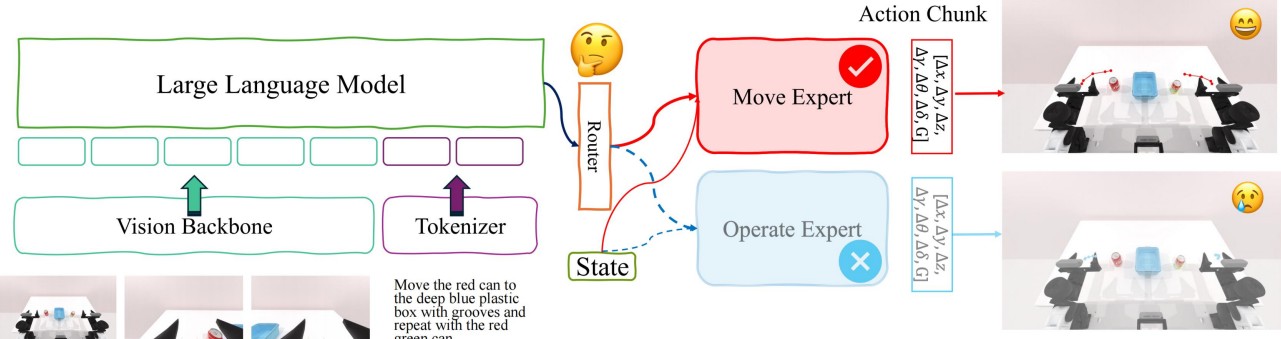

*Figure 3.* Overall model architecture. Dashed lines indicate data flow. And the rightmost images show the varying manipulation scales of the two experts. Only a single expert is active during actual inference.

2025) separates shared knowledge from domain specific experts under distribution shift. These results suggest that a structural decoupling can let specialized experts master move and operate without end to end compromise.

## 3. Preliminaries

### 3.1. Problem Formulation

We consider the problem of language-vision-conditioned robotic manipulation. Let $\tau = \{(C_t, \boldsymbol{a}_t)\}_{t=1}^{T}$ denote a demonstration trajectory, which represents a long-horizon sequence comprising distinct move and operate phases (and potentially repeated cycles). Here, $C_t = (I, O_t, S_t)$ represents the context tuple containing the language instruction $I$, visual observation $O_t$, and proprioceptive state $S_t$. The goal is to learn a model parameterized by $\theta$ that, given $C_t$, generates an action sequence $\boldsymbol{a}_t \in \mathbb{R}^{H \times d}$ given the context.

**Phase Decomposition.** Standard methods model the action distribution $p(\boldsymbol{a}_t \mid C_t)$ using a monolithic network. However, manipulation tasks can be naturally decomposed into two distinct phases with differing dynamics: move (reaching the target nearby) and operate (precise interaction). Optimizing a single set of parameters for these heterogeneous behaviors leads to gradient conflict and suboptimal performance in both regimes. To address this, we formulate action generation as a phase-conditional problem. We introduce a discrete latent variable $z \in \mathcal{Z} = \{\text{MOVE}, \text{OPERATE}\}$ and define the policy as a hard-switching model:

$$p(\boldsymbol{a}_t \mid C_t) = p(\boldsymbol{a}_t \mid C_t; \theta_{z_t}), \quad (1)$$

where $z_t \in \mathcal{Z}$ denotes the phase label at time $t$, determined by a high-level router. Introducing such a structural inductive bias not only aligns policy learning with the inherent two-phase structure of manipulation tasks, but also reduces optimization interference by enabling each expert $\theta_z$ to specialize in the dynamics of its corresponding phase.

### 3.2. Conditional Flow Matching

We parameterize the phase-specific experts using Conditional Flow Matching (CFM) (Lipman et al., 2023; Black et al., 2024). CFM learns a time-dependent vector field $v_\theta$ that transports a Gaussian prior $p_0 = \mathcal{N}(0, I)$ to the data distribution $p_1$. To distinguish from the trajectory step $t$, we denote the flow time as $\sigma \in [0, 1]$. Given the active label $z_t$, we define the flow state $x_\sigma = (1 - \sigma)x_0 + \sigma \boldsymbol{a}_t$. The velocity field is optimized by minimizing the regression loss and the $\theta_{\hat{z}_t}$ is the expert of $z_t$:

$$\mathcal{L}_{\text{FM}}(\theta_{\hat{z}}) = \mathbb{E}_{\sigma, x_0, \boldsymbol{a}_t} \left[ \| v_{\theta_{z_t}}(\sigma, x_\sigma, C_t) - (\boldsymbol{a}_t - x_0) \|_2^2 \right]. \quad (2)$$

During inference, given the predicted phase $z_t$, we synthesize the action sequence by solving the ordinary differential equation (ODE):

$$\boldsymbol{a}_t = x_0 + \int_0^1 v_{\theta_{\hat{z}_t}}(\sigma, x, C_t) \, d\sigma, \quad (3)$$

where the integral is approximated using a numerical solver.

## 4. Method

We propose **Move-Then-Operate**, a hierarchically gated policy formulated in Section 3. It is a neural architecture that introduces a structural inductive bias to explicitly decouple coarse relocation from fine-grained manipulation. The pipeline begins by encoding the multi-modal context $C_t$ via a shared VLM backbone, followed by a lightweight phase router that infers the latent label $z_t$. Conditioned on this routing signal, the system selectively activates one of two specialized Conditional Flow Matching experts $E_{\text{Move}}$ or $E_{\text{Operate}}$ to synthesize the action trajectory (see Fig. 3). By structurally isolating the parameter sets, our framework ensures that each expert optimizes a coherent vector field, effectively resolving the optimization interference inherent in monolithic baselines.

## 4.1. Dual-Expert Architecture

At each time step $t$, the policy conditions on observations $C_t$ to generate an action sequence $\boldsymbol{a}_t \in \mathbb{R}^{H \times d}$. We adopt the flow-matching parameterization described in Sec. 3. The model consists of a shared vision-language encoder and two distinct expert heads, denoted as $E_{\text{move}}$ and $E_{\text{operate}}$. These experts share the architecture of the base VLA model but maintain disjoint parameters. Such parameter isolation mitigates conflicting gradient updates between coarse transit and fine manipulation phases.

We introduce a latent variable $z_t \in \{\text{MOVE}, \text{OPERATE}\}$ to mediate expert selection via a hard-routing mechanism. Specifically, the global vector field is exclusively defined by the expert corresponding to the active phase—either $E_{\text{Move}}$ or $E_{\text{Operate}}$. Crucially, this expert selection remains invariant throughout the entire flow integration period $\sigma \in [0, 1]$ for each generated action sequence.

## 4.2. Routing Experts Per Action Chunk

We implement a chunk-level router that work on the high-level semantic features of the context. Unlike token-level MoE architectures that make routing decisions per token, our system selects a single expert for the entire control step $t$, ensuring temporal consistency across the generated action chunk.

First, the VLM backbone encodes the context $C_t$ into dense hidden states like

$$\mathbf{F}_t = \Phi_{\text{VLM}}(C_t), \ \mathbf{F}_t \in \mathbb{R}^{L \times D}. \tag{4}$$

To capture global scene dynamics, we compute a semantic summary $\mathbf{f}_t$ via masked global average pooling over $\mathbf{F}_t$. This design leverages the VLM's deep, pre-trained alignment of instruction and visual cues directly, avoiding reliance on shallow embeddings. The vector $\mathbf{f}_t$ drives a lightweight MLP router to predict the phase distribution:

$$p_\phi(z \mid C_t) = \text{Softmax}\big(\text{MLP}_{\text{router}}(\mathbf{f}_t)\big). \tag{5}$$

During inference, we select the active expert $\hat{z}_t$ via greedy decoding; the selected expert $E_{\hat{z}_t}$ then generates the action trajectory by attending to the shared representations $\mathbf{F}_t$.

## 4.3. Supervised Routing Learning and Inference

To enforce expert specialization, we employ a teacher-forcing strategy using a dataset $\mathcal{D} = \{(C_t, \boldsymbol{a}_t, y_t)\}$, where $y_t$ is the ground-truth phase. This decouples the router and expert optimization, preventing gradient conflict.

**Action Optimization.** We train the experts using the CFM objective. Crucially, we construct the velocity field using the ground-truth label $y_t$ rather than the router's prediction. The predicted velocity $v_{\text{pred}}$ is masked to activate only the

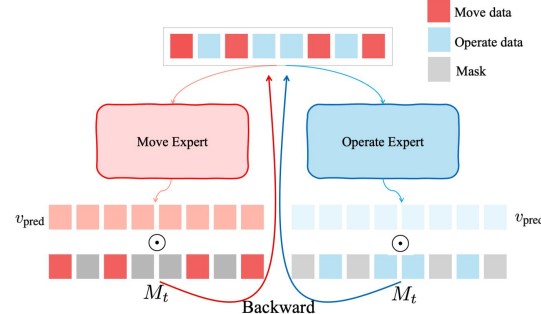

*Figure 4.* Data flow during the training process.

relevant expert:

$$v_{\text{pred}}(\sigma, x_\sigma, C_t) = \sum_{z \in \mathcal{Z}} \mathbb{I}[z = y_t] \cdot v_{\theta_{\hat{z}}}(\sigma, x_\sigma, C_t). \tag{6}$$

This formulation ensures $\nabla_{\theta_{\hat{z}}}$ is non-zero only for the matched expert, effectively orthogonalizing parameter updates. The loss is computed as the batch-level masked MSE against the target vector field $u_t = \boldsymbol{a}_t - x_0$:

$$\mathcal{L}_{\text{action}} = \mathbb{E}_{\sigma, x_0} \left[ \frac{\|M_t \odot (v_{\text{pred}} - u_t)\|_2^2}{\|M_t\|_1 + \epsilon} \right], \tag{7}$$

where $M_t$ handles variable batch size lengths (Fig. 4).

**Router Optimization and Inference.** The router is trained to mimic the ground-truth assignment by minimizing the cross-entropy loss on the pooled features $\mathbf{f}_t$:

$$\mathcal{L}_{\text{router}} = \text{CE}(p_\phi(\cdot \mid \mathbf{f}_t), y_t). \tag{8}$$

The total objective is $\mathcal{L}_{\text{total}} = \mathcal{L}_{\text{action}} + \lambda \mathcal{L}_{\text{router}}$. During inference, we strictly follow the routing pipeline: the active expert is selected via greedy decoding $z_t = \arg\max_z p_\phi(z \mid \mathbf{f}_t)$, and the action trajectory is synthesized by solving the ODE defined by $v_{\theta_{\hat{z}_t}}$.

## 4.4. Phase-Aware Auto-Labeling

We formulate the labeling task as a hierarchical temporal segmentation problem driven by a multimodal LLM $\mathcal{M}$. Given the video $V$ and instruction $I$, the model predicts a structured schedule $\mathcal{S}$ comprising $N$ consecutive subtasks. Formally, we define the output structure as:

$$\mathcal{S} = \big\{(T_i, \mathbf{p}_i)\big\}_{i=1}^N, \quad \text{where} \quad \mathbf{p}_i = (\phi_{i,1}, \dots, \phi_{i,m}). \tag{9}$$

Here, $T_i$ denotes the subtask interval, and $\mathbf{p}_i$ represents the sequence of atomic phases with types $\phi \in \{\text{MOVE}, \text{OPERATE}\}$.

**Structural Constraints and Assignment.** To ensure physical plausibility, we impose topological constraints on the phase sequence $\mathbf{p}_i$. Specifically, we restrict the decomposition depth such that $|\mathbf{p}_i| \leq 2$. In cases where a subtask

*Table 1.* Success rates on 8 representative RoboTwin2 tasks. All methods use 50 demonstrations per task and identical training steps. **Bold** indicates the best performance. We provide more visualization for each tasks in appendix.A

| Model | Click Alarmclock | Click Bell | Move Pillbottle pad | Place Bread Basket |
|---|---|---|---|---|
| ACT | 32% | 58% | 0% | 6% |
| RDT | 61% | 80% | 8% | 10% |
| $\pi_0$ | 63% | 44% | 21% | 17% |
| Ours | **88%** | **99%** | **37%** | **34%** |

| Model | Place Cans Plasticbox | Place Empty Cup | Place Burger Fries | Press Stapler |
|---|---|---|---|---|
| ACT | 16% | **61%** | 49% | 31% |
| RDT | 64% | 7% | 24% | 31% |
| $\pi_0$ | 34% | 37% | 80% | 62% |
| Ours | **79%** | 55% | **89%** | **70%** |

| **Overall Avg:** | ACT = 31.63%, | RDT = 35.63%, | $\pi_0$ = 44.75%, | Ours = **68.88%** |

bifurcates into two phases, they must possess distinct semantic types and strictly adhere to the chronological order observed in the video. This formulation allows for flexible temporal modeling without rigid templates. By aligning the validated intervals with the trajectory timestamps, we directly assign a semantic label $y_t \in \{\text{MOVE}, \text{OPERATE}\}$ to each control step.

**Validation and Self-Refinement.** We employ a deterministic validator $\mathcal{V}$ to enforce boundary continuity and structural validity. The inference process is modeled as an iterative feedback loop. At refinement step $r$, if the schedule $\mathcal{S}^{(r)}$ violates constraints, the validator generates a structured error description $\mathcal{E}(\mathcal{S}^{(r)})$. The model then updates its prediction conditioned on this feedback:

$$\mathcal{S}^{(r+1)} \leftarrow \mathcal{M}\big(V, I, \mathcal{S}^{(r)}, \mathcal{E}(\mathcal{S}^{(r)})\big). \qquad (10)$$

We terminate this process upon finding the first valid schedule $\mathcal{V}(\mathcal{S}) = \text{True}$ or reaching a maximum query budget, ensuring high-quality supervision for the downstream policy. Prompt and additional details are in the appendix.B.

## 5. Experiments

Decoupling move and operate into dedicated experts offers two key benefits. First, it introduces a structural inductive bias that enables specialized experts to capture action distributions tailored to their respective operational granularity. Second, it reduces learning complexity by mitigating optimization instability and gradient interference caused by conflicting objectives. Building on these advantages, we evaluate our approach through experiments that assess improvements in task performance, data efficiency, and training efficiency. Our experiments investigate three questions:

1. Does our model improve performance under a controlled data budget, using the same number of training samples and optimization steps?

2. Does our model has better data efficiency which lead

*Table 2.* Comparison against data-rich baselines. Models marked with * use $10\times$ more training data than ours. **Bold** indicates the best performance, and underlined indicates the second best.

| Task | $\pi_{0.5}^*$ | GO-1* | Ours |
|---|---|---|---|
| Click Alarmclock | **97%** | 95% | 91% |
| Click Bell | 75% | 98% | **99%** |
| Move Pillbottle pad | **33%** | 9% | 16% |
| Place Bread Basket | 48% | 47% | **49%** |
| Place Cans Plasticbox | 40% | **68%** | 16% |
| Place Empty Cup | **75%** | 44% | 29% |
| Place Burger Fries | 66% | **88%** | 61% |
| Press Stapler | 80% | 66% | **93%** |

competable with monolithic baselines trained on significantly larger datasets?

3. Does our model has better training efficiency, which means we can achieve same or better performance with less training steps?

### 5.1. Experimental Setup

**Initialization and Architecture.** We implement our model based on $\pi_0$ (Black et al., 2024) and initialize our model from the pre-trained $\pi_0$-base checkpoint. To maintain parameter efficiency, we inject distinct Low-Rank Adapters (LoRA) (Hu et al., 2022) for the two experts and vision-language backbone. This design allows regime-specific dynamics learning while preserving the generalized representation of the backbone.

**Benchmark and Detail.** We conduct evaluations on RoboTwin2(Chen et al., 2025), focusing on 8 tasks that involve diverse contact dynamics and object geometries (e.g., *Click Alarmclock*, *Place Bread Basket*). Following original training method, we first train the model across all 50 tasks, with each task containing 50 demonstration trajectories collected in clean scenes. During this process,

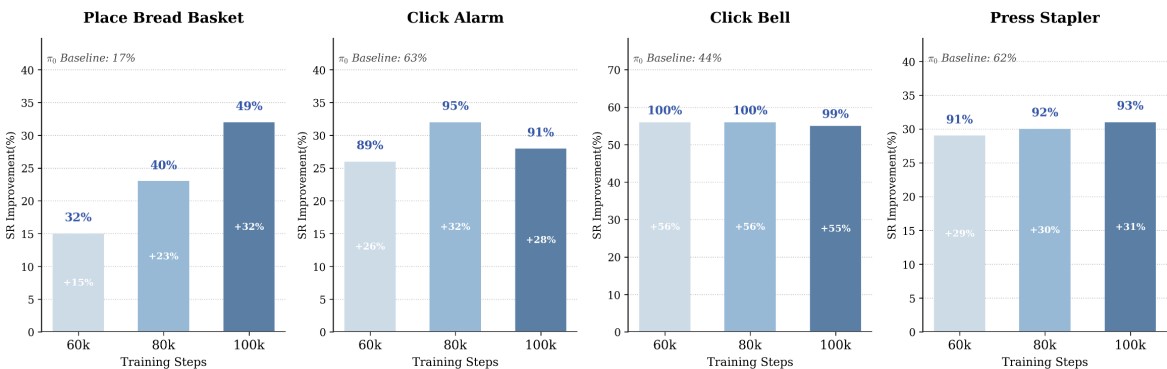

*Figure 5.* Success rates of our model at 60k, 80k, and 100k training steps during the multi-task pre-training phase. $\pi_0$ Baseline represents the reported performance of the monolithic baseline.

we maintain an equal sampling ratio between the move and operate phases. This training stage proceeds for 100,000 steps. Subsequently, we fine-tune the model individually on 8 specific tasks using the same trajectory data, training each for an additional 20,000 steps. In evaluation, We report the average success rate over 100 trials per task in clean scenes.

## 5.2. Results and Analysis

### 5.2.1. MAIN RESULTS

Table 1 summarizes the evaluation on the RoboTwin2 benchmark with a constrained budget of 50 demonstrations per task. Our method achieves an average success rate of 68.9%, outperforming the $\pi_0$ baseline by +24.1%. Notably, the performance gains are most distinct in tasks requiring high-precision actuation. For contact-critical tasks such as *Click Bell* and *Press Stapler*, our dual-expert architecture improves success rates by 55% and 8%, respectively, compared to the next best method. We also observe consistent improvements in long-horizon transport tasks like *Place Cans*, where our method improves the success rate from 64% to 79%, suggesting that explicitly decoupling coarse relocation from fine manipulation effectively mitigates optimization interference.

### 5.2.2. DATA EFFICIENCY

Table 2 investigates whether our structured policy can compete with monolithic baselines trained on significantly larger datasets. Our model is trained only on 50 tasks with 50 demonstrations per task, using a total budget of 100,000 training steps (*no task-specific fine-tuning is performed in this part*). We benchmark against $\pi_{0.5}^*$ (Intelligence et al., 2025; Bi et al., 2025) and GO-1* (AgiBot-World-Contributors et al., 2025; Bi et al., 2025), which are trained on an order-of-magnitude more data ($10\times$ our data budget). Despite this substantial disparity in data scale, our method remains highly competitive and even achieves superior performance in specific domains.

The results reveal a clear distinction between tasks where performance is primarily governed by precise physical interaction and those requiring broad visual generalization. In contact-rich tasks such as *Press Stapler* and *Click Bell*, our method outperforms $\pi_{0.5}^*$ by margins of +13% and +24%, respectively. These tasks rely heavily on the precise execution of end-effector rather than visual recognition. By isolating the operate phase, our dual-expert architecture enables the model to learn fine-grained manipulation policies from limited samples. This suggests that for complex manipulation skills, architectural specialization can be a more effective driver of performance than simply scaling up dataset size.

Conversely, the data gap emerges in visually diverse tasks like *Place Cans Plasticbox*, which involve varied object geometries and orientations requiring robust visual representations. Baselines, with $10\times$ more visual exposure, generalize better across spatial variations. Our model, limited by the small dataset's visual diversity, sometimes struggles on unseen configurations despite learning correct motion primitives. Nevertheless, our method achieves strong performance with only one-tenth the data, demonstrating both effectiveness and data efficiency.

### 5.2.3. TRAINING EFFICIENCY

To evaluate the optimization efficiency of our framework, we analyze the learning curves across two critical stages: multi-task pre-training and task-specific fine-tuning.

**Multi-task Training Performance.** We first evaluate the success rates at different checkpoints (60k, 80k, and 100k steps) during training across 50 tasks. As illustrated in Figure 5, our model demonstrates rapid convergence. For instance, in contact-intensive tasks such as *Click Bell* (100%) and *Click Alarm* (89%), our model achieves near-peak performance at only 60k steps, surpassing the final performance of the $\pi_0$ baseline which is trained for 100k steps and fine-

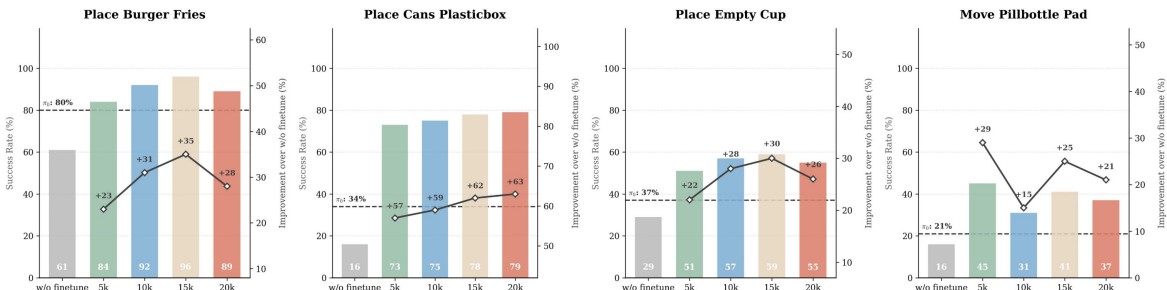

*Figure 6.* Success rates during task-specific fine-tuning at 5k, 10k, 15k, and 20k steps. The 'w/o finetune' column denotes the performance of the pre-trained multi-task model.

tuned for 20k steps. Similar efficiency is observed in *Press Stapler*, which reaches $91\%$ early in the training process. This accelerated learning suggests that the decoupled Move-Then-Operate architecture simplifies the joint optimization of transport and manipulation dynamics, allowing the policy to learn phase-critical behaviors more effectively than monolithic approaches.

**Fine-tuning Convergence.** We further investigate the efficiency of task-specific performance by fine-tuning the multi-task trained checkpoint on individual tasks. Figure 6 shows the success rates at 5k, 10k, 15k, and 20k steps. The results demonstrate rapid convergence. In *Place Cans Plasticbox*, the success rate improves from $16\%$ to $73\%$ within the first 5k steps. In the *Place Burger Fries* task, the model reaches a peak success rate of $96\%$ at 15k steps.

While most tasks reach their performance peak between 10k and 15k steps, we observe that performance tends to saturate or exhibit minor variation by 20k steps in certain tasks, such as *Move Pillbottle Pad*. All observations indicate that the decoupled experts can rapidly specialize to task-specific dynamics within a limited fine-tuning budget, highlighting the practical value of our framework.

### 5.3. Ablation Study

To investigate the necessity of accurate phase routing and the degree of expert specialization, we compare our method against two counterfactual strategies: Random Selection, where the active expert is chosen stochastically for each action chunk, and Reversal Selection, an adversarial setting where phase assignment is explicitly inverted. The results in Table 3 validate the critical role of our routing mechanism, as replacing the learned router with Random Selection causes a precipitous drop in the average success rate from $68.88\%$ to $25.63\%$, while the adversarial Reversal Selection further degrades performance to just $8.88\%$. This result confirms that our experts specialize in distinct, incompatible behaviors. Consequently, using the wrong expert causes severe conflict failure, rather than just reduced performance.

*Table 3.* The task success rates for various selection strategies are presented. Random denotes selections made independently of the router's predictions. Reversal refers to taking the inverse of the router's decisions, while Original represents the results based on the router's initial predictions

| Task | Random selection | Reversal selection | Original selection |
|---|---|---|---|
| Click Alarmclock | 46% | 15% | 88% |
| Click Bell | 41% | 7% | 99% |
| Move Pillbottle pad | 4% | 0% | 37% |
| Place Bread Basket | 14% | 11% | 34% |
| Place Cans Plasticbox | 8% | 2% | 79% |
| Place Empty Cup | 11% | 1% | 55% |
| Place Burger Fries | 40% | 0% | 89% |
| Press Stapler | 41% | 35% | 70% |
| Average | 25.63% | 8.88% | 68.88% |

The impact of routing errors varies significantly by task complexity. For short-horizon tasks like Press Stapler, the Reversal strategy retains a 35% success rate as the experts may partially generalize across simple motions, but for long-horizon or high-precision tasks, the failure is absolute. In Place Cans Plasticbox and Move Pillbottle Pad, success rates under Reversal plummet to 2% and 0% respectively because the Move expert lacks the fine-grained dexterity for placement while the Operate expert lacks the velocity and amplitude for transport, causing cumulative trajectory errors that make task completion impossible.

## 6. Conclusion

We present **Move-Then-Operate**, a human-inspired VLA framework that explicitly disentangles coarse relocation from contact-rich manipulation via a phase-routed dual-expert policy, reducing optimization interference between heterogeneous control regimes. Our key idea is to enforce phase-wise specialization under imitation-heavy supervision, supported by an automated phase labeling pipeline with lightweight contextual cues, so that abundant move segments do not drown out the learning of fine-grained operational skills. Experiments on RoboTwin2 show that this decoupling yields substantial gains over a monolithic $\pi_0$ baseline (*e.g.*, **68.9%** average success with **+24.1%** absolute improvement) and improved data and training efficiency. We expect phase-disentangled VLA designs to serve as a

scalable path toward robust, high-precision robotic manipulation under practical data and compute constraints.

## Impact Statement

This paper presents work whose goal is to advance the field of Machine Learning. There are many potential societal consequences of our work, none which we feel must be specifically highlighted here.

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

# A. Visualization

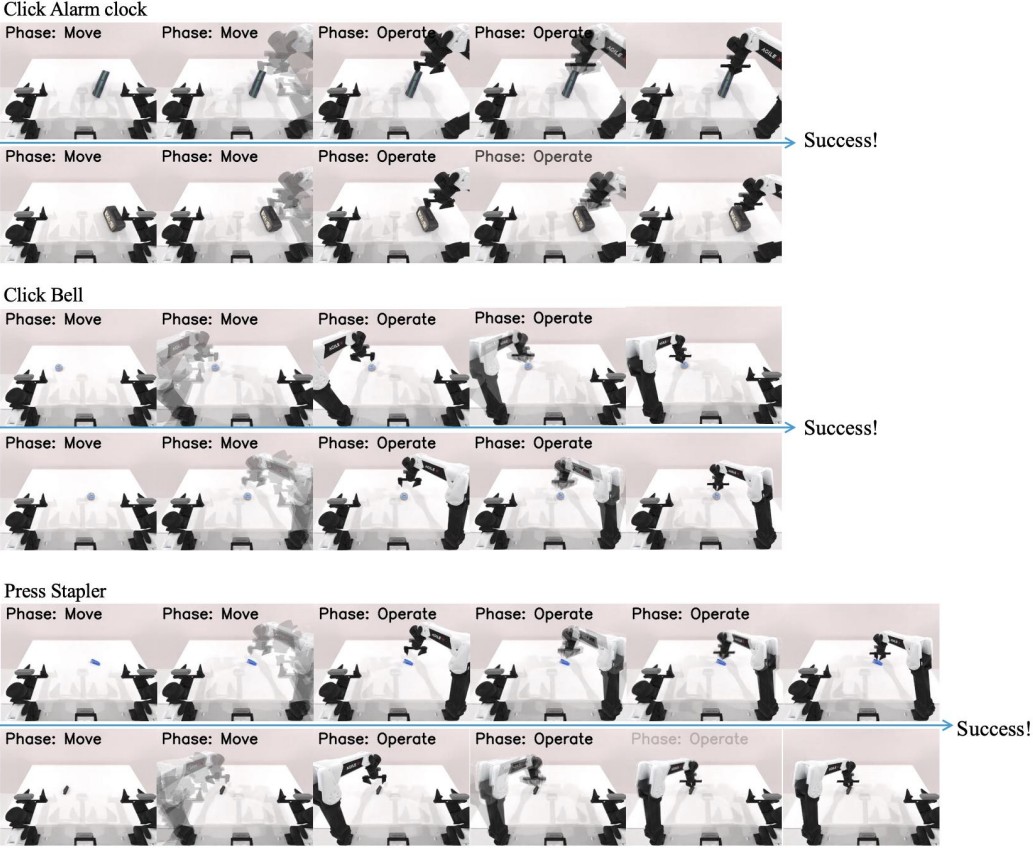

*Figure 7.* Visualization of task click alarmclock, click bell and press stapler.

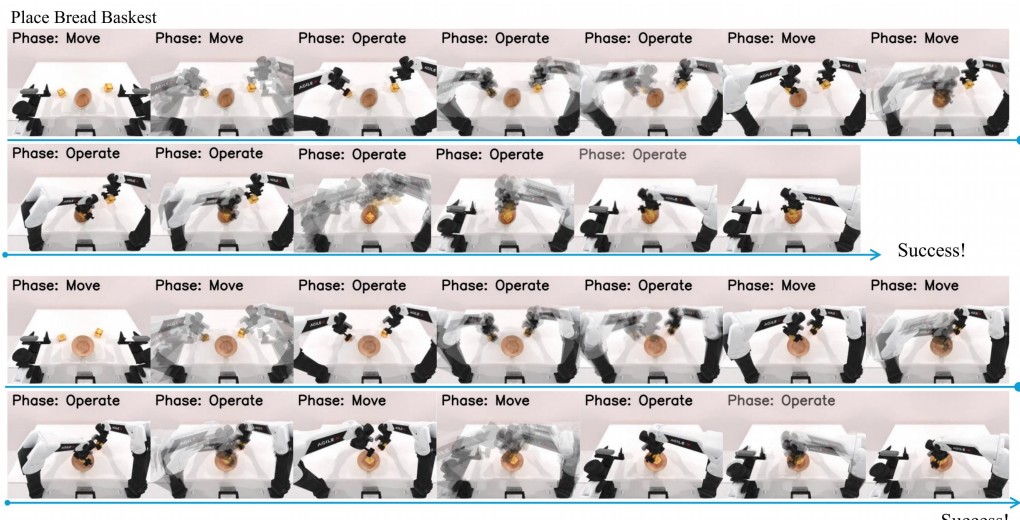

*Figure 8.* Visualization of place bread basket.

We visualize the entire execution process for task in the test set in Fig. 7, 8, 9 and 10. Sampling is conducted at a consistent frequency and interval, where overlapping frames represent the full progression of the sampling window. Upon a state transition, we display a static image captured at the initial timestamp.

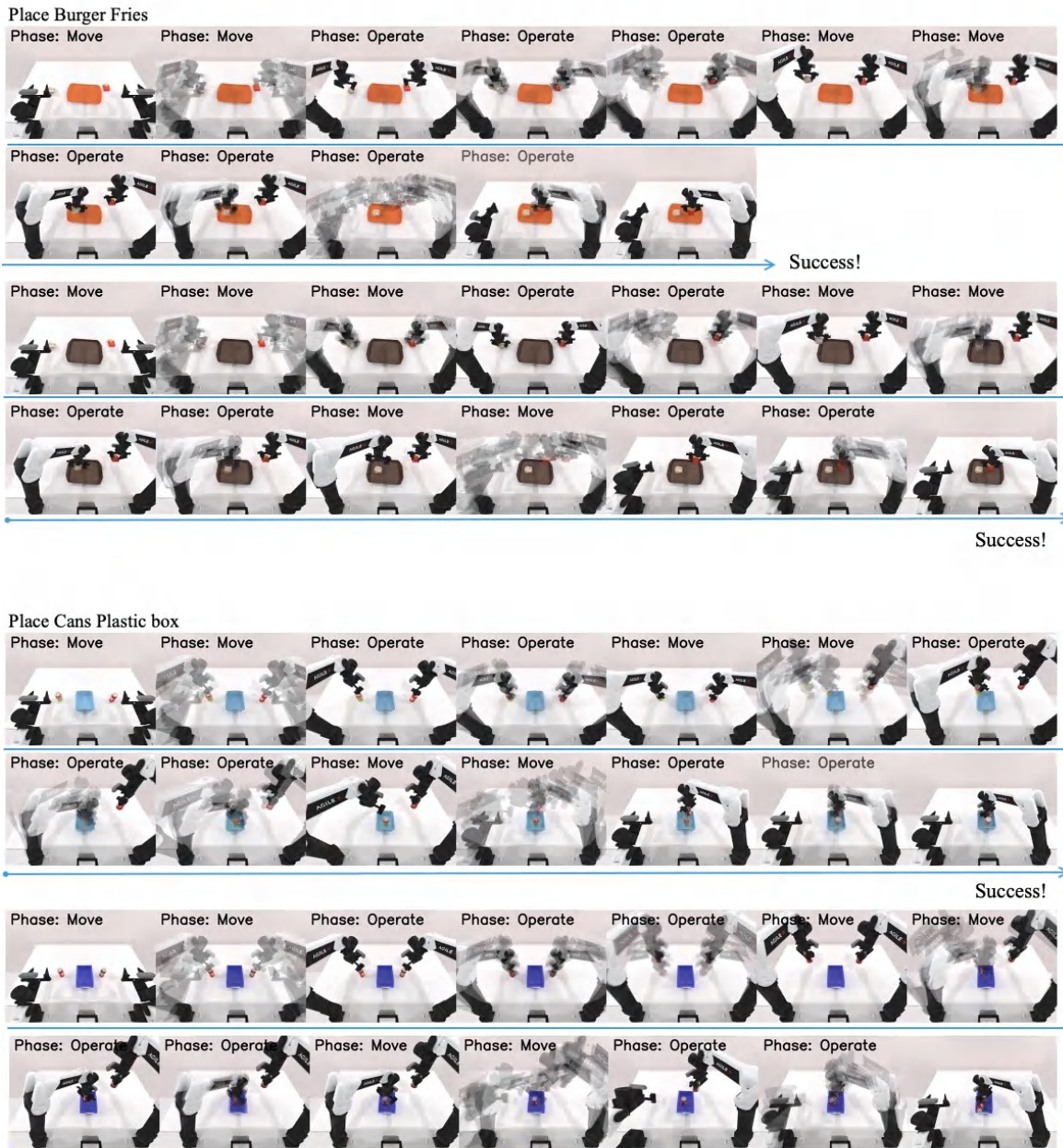

*Figure 9.* Visualization of place burger fries and place cans plasticbox.

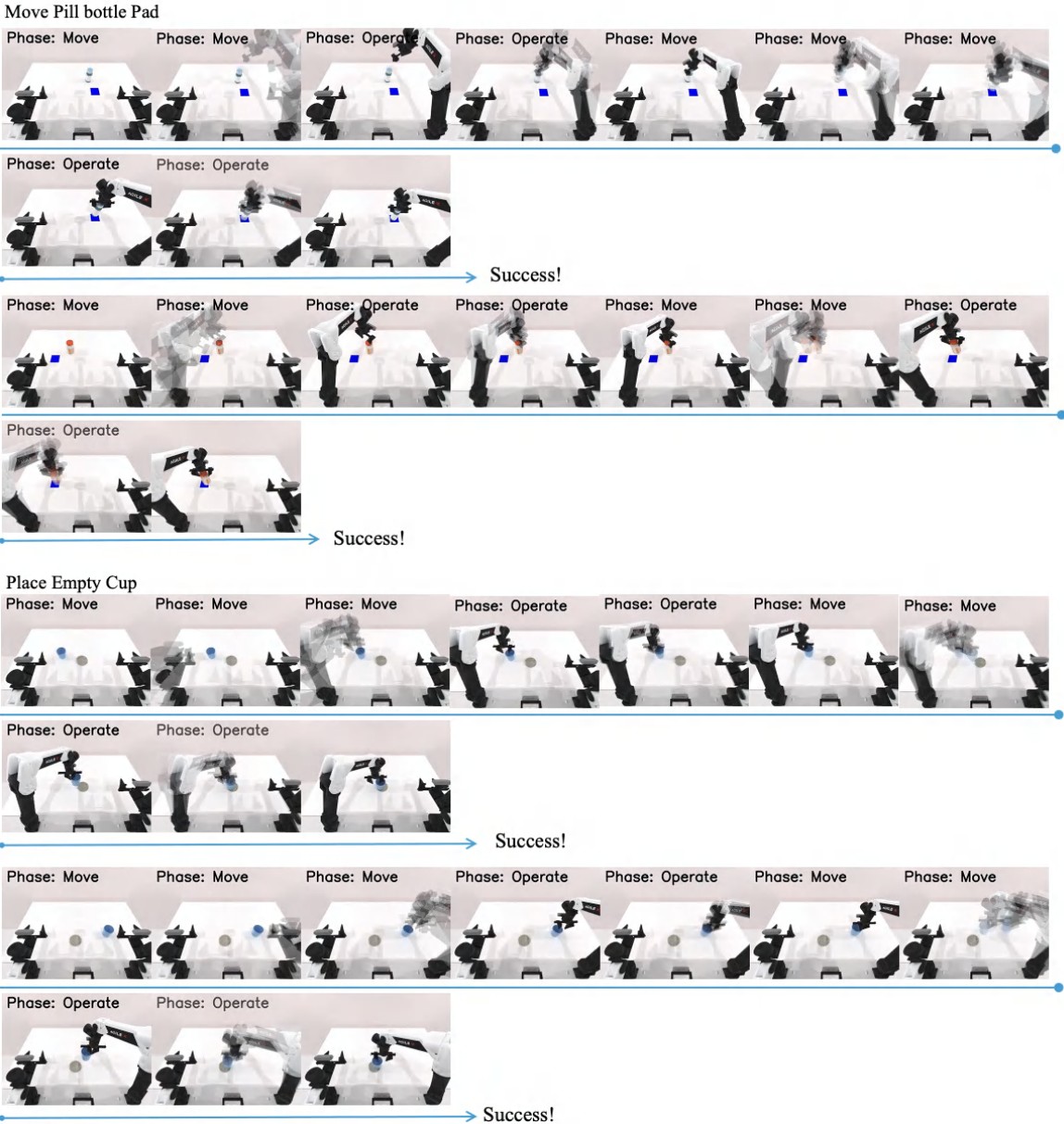

*Figure 10.* Visualization of move pillbottle pad and place empty cup.

---

**System Prompt (Initial Input)**

**Role:** You are an expert robotic manipulation annotator.
**Context:** Instruction: `<Instruction String>`. Video cadence: 30 FPS. Total frames: `<N>`.
**Task Requirements:**

1. Segment the entire video into consecutive subtasks that fully cover [0, total_frames-1] without gaps/overlaps.

2. For each subtask, output phases (temporal slices within the subtask):

   - `phase_type` in {`move`, `operate`}.
   - A subtask may have ONE phase OR TWO phases (exactly one move and one operate).
   - Do NOT repeat the same `phase_type` within a subtask. If you observe another movement, start a NEW subtask.

3. Identify the `primary_arm` and give a concise English description.

4. Predict normalized coordinates for `target_object_axis` and gripper ends.

5. Ensure the entire video contains AT LEAST ONE `move` phase.

**Output Schema:** Output STRICTLY a JSON array. No extra text.

```
[
  {
    "subtask": 1,
    "subtask_description": "...",
    "primary_arm": "left/right/both/unknown",
    "phases": [
      { "phase_type": "move", "start_frame_idx": 0, "end_frame_idx": 45, ... },
      { "phase_type": "operate", "start_frame_idx": 46, "end_frame_idx": 120, ... }
    ],
    ...
  }
]
```

*Figure 11.* The foundational prompt used to instruct the MLLM for zero-shot phase segmentation.

---

**Refinement Prompt (Triggered on Validation Failure)**

**Trigger Condition:** The Validator $\mathcal{V}$ detects a structural or physical violation in the previous output.
**Injected User Message:**
`Previous attempt issues:` **`<Error Log from Validator>`**.
Fix by ensuring:

1. Do not duplicate a `phase_type` inside a subtask; instead start a new subtask for the extra action.

2. Each subtask has 1 phase (move/operate) or exactly 2 (one move + one operate) in real temporal order.

3. The entire video contains at least one `move` phase.

Return JSON array only.

*Figure 12.* The dynamic feedback prompt. The `<Error Log>` is replaced by the actual exception message (e.g., "Subtask 2 end frame exceeds total frames"), guiding the model to fix the specific issue.

## B. Detail of Auto-Labeling

During the annotation process, we utilized Seed 1.6 Vision as the backbone model. We applied a sampling rate of 5 fps for all samples, with each trajectory capped at a maximum of 64 frames. This configuration is sufficient to cover the longest videos in our training dataset. We employ a two-stage prompting strategy: an initial generation phase followed by an

iterative refinement phase triggered by validation errors. The MLLM is initialized with a detailed system instruction that defines the temporal segmentation task, the physical constraints of the robot (e.g., phase types), and the required JSON output schema like Fig.11.

When the generated JSON fails our deterministic validation checks (e.g., overlapping timestamps, missing phases, or invalid transitions), we do not discard the result. Instead, we feed the specific error message back to the model as a new user prompt to trigger a self-correction like Fig. 12.

