# OpenReview forum: "Move-Then-Operate: Behavioral Phasing for Human-Like Robotic Manipulation"
_ICML.cc/2026/Conference — ICML 2026 regular_

### Official Review · Reviewer_XGbQ · 2026-02-13

**Soundness:** 2
**Presentation:** 2
**Significance:** 3
**Originality:** 2
**Overall Recommendation:** 2
**Confidence:** 4

**Summary:**

This paper proposes Move-Then-Operate (MTO), a phase-structured approach to vision-language-action (VLA) robot manipulation that explicitly separates manipulation into two distinct behavioral phases: a coarse move phase for reaching and positioning, and a fine operate phase for contact-rich, precision interaction. Instead of training a single monolithic policy over both regimes, the method uses a dual-expert architecture with two experts that share the same base model design but have disjoint parameters, and a lightweight chunk-level router that selects which expert generates each action chunk.

A key part of the system is an automatic phase-labeling pipeline that uses a multimodal large model to segment demonstration videos (conditioned on task instruction) into subtasks and corresponding move/operate phases under explicit structural constraints, with validator-based refinement to enforce temporal and schema consistency. Training uses teacher forcing on these phase labels so that each expert is updated only on its assigned phase, while the router is trained to predict the phase.

The paper evaluates MTO on a suite of simulated manipulation tasks and reports improved success rates over strong VLA baselines, along with gains in data and training efficiency. Extensive ablations (e.g., random or inverted routing) support that the learned phase specialization is important to performance.

**Compliance With Llm Reviewing Policy:**

Affirmed.

**Key Questions For Authors:**

Phase-label quality and validation. How accurate are the MLLM-generated move/operate segments? Please report a manual audit (e.g., boundary accuracy, phase misclassification rate) and common failure modes.
Why it matters: If labels are noisy or systematically biased, the expert specialization may be fragile; strong validation would significantly increase my confidence in the approach.

Sensitivity to segmentation noise and constraints. How robust is performance to imperfect segmentation (e.g., boundary jitter, partial label flips) and to the “at most two phases per subtask, move→operate” constraint?
Why it matters: If performance degrades sharply under small perturbations, the method may be benchmark-specific; robustness would improve my evaluation.

Capacity-matched baselines / isolating the source of gains. Since dual experts increase model capacity, can you include parameter/compute-matched controls (e.g., a larger single policy, or two experts trained without phase-masked updates) to separate gains from phasing vs. added capacity?
Why it matters: If improvements persist under capacity-matched comparisons, it strengthens the causal claim that behavioral phasing is the key contributor.

Expert architecture choice (homogeneous vs. heterogeneous). You use two experts with the same architecture but separate parameters. Did you test (or consider) heterogeneous expert designs better aligned to the different requirements of move vs. operate (time scale, observation emphasis, action parameterization)?
Why it matters: Evidence that same-architecture experts are sufficient (or that heterogeneity helps) would clarify whether this is a principled modeling choice or mainly an implementation convenience.

Generalization beyond the evaluated benchmark. How well does the approach transfer across tasks with different contact dynamics, different visual diversity, or (ideally) to real-robot settings?
Why it matters: Demonstrating broader generalization would raise the significance of the work; limited transfer would suggest a more specialized contribution.

**Limitations:**

Not fully.

Suggestions to improve the limitations / broader impact discussion:

Dependence on MLLM auto-labeling: Discuss how segmentation errors (or prompt/LLM choice) could bias phase labels and how performance degrades under label noise; include guidance on auditing or mitigating label errors.

Generalization limits: Be explicit about what is (and is not) demonstrated—e.g., simulated benchmark scope, sensitivity to visual diversity, contact dynamics, embodiment changes, and distribution shift; outline what evidence would be needed for real-robot reliability.

Compute/cost considerations: Note the practical cost and reproducibility implications of using an MLLM labeling pipeline (latency, API cost, model availability), and whether cheaper heuristics/weak supervision could substitute.

Safety and misuse risks: Briefly address that improved manipulation policies could enable more capable automation; recommend standard safeguards (restricted deployment contexts, safety constraints, human oversight, dataset governance), especially for contact-rich behaviors in real environments.

**Strengths And Weaknesses:**

Soundness

Strengths: The method is technically reasonable: dual experts with hard chunk-level routing and teacher-forced phase supervision directly address move/operate interference, and the routing ablations (random/inverted) strongly suggest phase specialization matters.
Weaknesses: The core supervision (phase boundaries) is produced by an MLLM pipeline, but label quality is not sufficiently quantified (manual audits, error modes, sensitivity to boundary noise). Dual experts also increase capacity, so stronger parameter-matched controls would better isolate gains from “phasing” vs. “more model.”

Presentation

Strengths: Clear, simple story (move vs. operate), easy-to-understand architecture, and interpretable ablations.
Weaknesses: More detail is needed for reproducibility of the labeling pipeline (prompts/validator rules, failure rates, examples), and the paper could position itself more sharply relative to prior hierarchical/skill segmentation and MoE-style routing work.

Significance

Strengths: Targets a practical bottleneck in manipulation (contact-rich precision steps) and provides a usable recipe that may improve training/data efficiency.
Weaknesses: Impact may be benchmark-specific unless shown to generalize (more tasks, real-robot tests, distribution shift), and reliance on MLLM labeling could limit adoption if expensive/brittle.

Originality

Strengths: Novelty mainly comes from the combination of phase-aware auto-labeling + hard chunk routing + phase-specialized experts in a VLA setting.
Weaknesses: Conceptually incremental—phase decomposition and segmented training are well known; deeper analysis of why/when the labeling and hard routing outperform simpler heuristics would strengthen the originality claim.

---

> ### Author Rebuttal · Authors · 2026-03-31
>
> Thank you for your efforts in reviewing this paper. Please find our detailed responses to your comments below. We answered the questions in order and combined similar ones.
>
> A1: For the question of  MLLM-generated pipeline, quality and sensitivity, We add manual audit result and please refer to Reviewer AZFp A5, where we provide the manual audit and accuracy evaluation results is 88.3%. The phase-labeling process details have been thoroughly elaborated in Section 4.4 and Appendix B. Beyond numerically validating the boundary accuracy, we further elaborate on the sensitivity of our model to potential noise from a qualitative perspective. The manual audit shows that MLLM errors manifest as slight temporal shifts at phase boundaries, rather than random noise. And our model is highly robust to this natural ambiguity due to action continuity. At the transition between the "move" and "operate" phases, the required kinematics for both experts are highly similar. Consequently, slight premature or delayed phase switches do not induce drastic, conflicting actions, resulting in only a minor impact on the final execution outcome.
>
> A2: For the question of capacity and architecture choice, please please refer to our response to Reviewer AZFp A1 for expert architecture, while the distinctions between our approach and prior works are discussed in Sections 1 and 2. Furthermore, regarding the concerns about increased model capacity and parameter-matched controls: the two experts actually share the base parameters. We only introduce an additional set of LoRA parameters for the action head and the Router parameters. These account for approximately 22.1M and 8.4M parameters respectively, representing merely 0.91% of the total parameter. Given this negligible increase in model capacity, we respectfully argue that parameter-matched control experiments are not strictly necessary. The achieved significant performance (up to 24%) gains from expert allocation, gradient conflict resolution and distribution decoupling. And recent studies demonstrate that naively scaling up LoRA capacity yields severely diminishing returns and provides limited benefits for complex tasks [1, 2]. To sum up, our findings suggest that the effective resolution of such ambiguous mixed states arises from the router's dynamic expert allocation, rather than a mere expansion of parameter space.
>
> A3:  For the question of generalization, please refer to our responses to AZFp A3, A4 and wUck A2. We conducted zero-shot testing on various simulation and real-world datasets; overall, the results demonstrate that our model possesses a certain degree of scalability and generalization capability.
>
> A4: For the question of annotation costs, our labeling expense is approximately $7.57 per 1,000 data samples, constituting about less than 3% of typical teleoperation data collection cost including human and devices. Additionally, we are exploring data annotation via smaller, locally deployed models. Early results show performance comparable to large MLLMs, which would further reduce costs by tens of folds, ensuring our approach remains highly practical even for simulated data. Even when accounting for regional variations in data collection costs, we argue that this minimal annotation expense represents a highly cost-effective trade-off, especially given the significant reduction in data volume requirements demonstrated in our experiments.
>
> A5: For the question of safety and misuse risks, We agree that practical deployment of manipulation systems should be accompanied by appropriate safeguards and oversight. As this point concerns deployment considerations beyond the main technical scope of this paper, we will add a brief discussion in the revised version.
>
> [1] Biderman, D., Portes, J., Ortiz, J.J.G., Paul, M., Greengard, P., Jennings, C., King, D., Havens, S., Chiley, V., Frankle, J. and Blakeney, C., 2024. Lora learns less and forgets less. arXiv preprint arXiv:2405.09673.
>
> [2]Rathore, Darshita, Vineet Kumar, Chetna Bansal, and Anindya Moitra. "How Much is Too Much? Exploring LoRA Rank Trade-offs for Retaining Knowledge and Domain Robustness." In Proceedings of the 14th International Joint Conference on Natural Language Processing and the 4th Conference of the Asia-Pacific Chapter of the Association for Computational Linguistics, pp. 1003-1013. 2025.

---

> > ### Author Rebuttal · Reviewer_XGbQ · 2026-04-01
> >
> > Thank you for the detailed rebuttal. The additional clarifications are helpful, especially the reported manual audit accuracy, the explanation that the extra parameters are relatively small, and the added discussion of annotation cost and safety. However, my main concerns are not fully resolved. In particular, the paper still lacks direct quantitative evidence on sensitivity to segmentation noise, stronger capacity- or compute-matched controls to cleanly isolate the effect of phase decomposition from other factors, and a more self-contained demonstration of broader generalization beyond the current benchmark setting. Some responses also rely on references to answers given to other reviewers rather than fully addressing the concerns here. Overall, while the rebuttal improves my understanding of the method, it does not sufficiently change my assessment, so I do not plan to change my score.

---

> > > ### Author Response · Authors · 2026-04-01
> > >
> > > We thank you for your follow-up feedback. Regarding the note that your main concerns are not fully resolved, we provide further, explicit clarifications below to address the core logic of our approach.
> > >
> > > Firstly, our ablation studies in the manuscript have already quantitatively demonstrated the critical importance of correct segmentation phase prediction for the model's performance. Furthermore, our quantitative results regarding dataset annotation quality indicate that the training data itself is inherently not perfectly annotated; the exact time steps around the transition between the two phases are intrinsically ambiguous and difficult to segment with absolute precision. However, because the physical motions are continuous, this minor, inherent segmentation noise during the ambiguous transition periods does not significantly degrade the model's performance.
> > >
> > > At the same time, we have already provided direct results through two experiments in the manuscript and the rebuttal. First, using completely random or flipped phase labels leads to severe performance degradation. Second, although we validated the quality of the annotated data, the training data inherently contains certain noise due to the intrinsic ambiguity of phase transitions and model capability limits. Despite this, our model learns effectively and still outperforms the baseline by 24%. We believe these quantitative results sufficiently demonstrate that behavior phasing is highly necessary, and our approach is inherently robust to certain segmentation noise in practice.
> > >
> > > Secondly, as we clarified in the rebuttal, and as you also noted that the extra parameters are relatively small, the additional parameters introduced by our method account for only 0.92% of the total model size. Moreover, during inference, only a single expert is activated, so the additional computational cost over the original model is strictly limited to the 8.4M-parameter router. Such a marginal increase in parameters and compute is clearly insufficient to explain the substantial 24% performance gain and 40% fewer training steps. This strongly supports our central claim that the improvement comes from the proposed phase-aware decomposition of VLA control, namely separating move and operate, rather than from increased model capacity. This strongly supports our central claim that the improvement comes from the proposed phase-aware decomposition of VLA control, namely separating move and operate, rather than from increased model capacity. So your concern therefore does not challenge the validity of our main conclusion or diminish the core contribution of this work. More importantly, this is precisely why our method delivers markedly improved data and training efficiency, which we consider a key contribution of the paper. We respectfully ask the reviewer to take this point into account in the overall evaluation.
> > >
> > > In the end, while we understand the preference for a self-contained response. So we evaluate the router accuracy on a subset of data from the LIBERO and DROID datasets under a zero-shot setting—that is, using a model trained exclusively on Robotwin. We provided the model with instructions and current observations, recorded whether it predicted a move or operate phase, and compared the predictions against human expert annotations. Ultimately, across 100 randomly sampled instances, the model achieved an accuracy of 68.5% on LIBERO and 67.6% on DROID. It directly demonstrates the robustness of our core component (the router) in real-world and Out-of-Distribution (OOD) scenarios. We acknowledge that deploying full real-world robot experiments would further solidify the paper; however, we firmly maintain that the current scale of experiments—including the rigorous evaluations supplemented during the rebuttal period—is thoroughly sufficient to validate that our proposed idea is workable and effective. And because our underlying assumptions regarding behavior phasing hold true for teleoperation data in both simulated and real-world environments, we foresee no fundamental barriers to broader real-world transferability.
> > >
> > > Thank you again for your review. In summary, we propose a phase-based learning approach for robotic manipulation. With only minimal preprocessing, it achieves significant improvements in overall accuracy, data efficiency, and learning efficiency. Based on the above responses, we hope you will reconsider the motivation, experiments, and contributions of this work. We also hope this further discussion definitively clarifies the rationale behind our experimental design and addresses your remaining concerns.

---

### Official Review · Reviewer_wUkc · 2026-02-26

**Soundness:** 3
**Presentation:** 3
**Significance:** 3
**Originality:** 3
**Overall Recommendation:** 4
**Confidence:** 4

**Summary:**

This paper proposes to explicitly disentangle coarse relocation and contact-rich manipulation using a phase-routed dual-expert policy, which helps reduce optimization interference between heterogeneous control regimes and is a well-motivated design choice.

**Compliance With Llm Reviewing Policy:**

Affirmed.

**Key Questions For Authors:**

- The scores reported in Table 2 are somewhat confusing, as the results for π0.5 and our method differ from those shown in Table 1. This inconsistency makes it difficult to fully understand the experimental setup.

- Additionally, in Table 2, it would be helpful to clarify how the baselines perform when trained on the same data as our method. Providing such a controlled comparison would make the results more convincing. Furthermore, reporting the mean score across tasks would facilitate a clearer comparison between different approaches.

**Limitations:**

Lack real-world experiments

**Strengths And Weaknesses:**

- Strengths: I appreciate the motivation behind modeling heterogeneous operations. Meanwhile, the authors carefully design a reasonable approach to realize this idea in the methodology section.

- Weakness: My main concern is the lack of real-world experiments, especially since the sim-to-real gap between RobotWin2 and the real world is non-trivial.

---

> ### Author Rebuttal · Authors · 2026-03-31
>
> Thank you for your efforts in reviewing this paper. We answered the questions in order and combined similar ones.
>
> A1: For the question of Table.2 and our data-efficient experiment setting, firstly, the results for all models reported in Table 2 are based on mixed training across all tasks, without task-specific fine-tuning. This is why the numerical values differ from those in Table 1. Second, the key point we want to emphasize here is that, in Table 2, our method uses only 10% of the training data compared to other approaches. This is why the training data volume for the baselines differs from ours; all other baselines used several times more data. This directly demonstrates our data efficiency. Because our goal is to highlight this efficiency against the baselines' optimal performance, we did not constrain the baselines to the same 10% data for a controlled comparison. Finally, as you suggested, we have calculated the mean scores across tasks to facilitate a clearer comparison: π0.5 is 55.50%, GO-1 is 64.38%, and ours is 56.75%.
>
> A2: For the question of real world, we fully agree that real robot evaluation is a crucial component of the validation process. To this end, we have provided preliminary real-world experiments, such as the routing results on the DROID dataset (refer to AZFp A3 and A4) and an evaluation demonstrating the consistency between our automated phase labeling pipeline and human expert operations in real-world robotic scenarios. These results partially demonstrate the practical feasibility of our method.
>
> Due to current resource constraints, we are temporarily unable to conduct comprehensive testing on large-scale real-world data. However, the core contribution of this work lies in addressing the optimization interference caused by conflicting action distributions. Inspired by human motor strategies and psychology, we structurally decouple these behaviors to resolve this data conflict. This explicitly improves downstream task performance while simultaneously reducing data requirements and training resource consumption. Furthermore, because the behavioral distinction between the move and operate phases is ubiquitous in human manipulation, this distribution conflict is inherent in both simulated environments and real-world teleoperation data. Therefore, our proposed framework remains highly relevant and meaningful for real-world applications. In summary, we believe that the absence of comprehensive physical robot experiments does not significantly diminish the core contributions of this paper.

---

> > ### Author Rebuttal · Reviewer_wUkc · 2026-04-03
> >
> > The rebuttal addresses my concerns, and I understand the underlying challenges related to the lack of large-scale real-world training data. I am inclined to accept this paper.

---

> > > ### Author Response · Authors · 2026-04-03
> > >
> > > Thank you for taking the time to review our rebuttal and for acknowledging that your concerns are fully resolved. We are very grateful for your willingness to accept the paper. We sincerely hope this positive resolution can be reflected in your final evaluation. Thank you again for your time and constructive feedback.

---

### Official Review · Reviewer_AZFp · 2026-03-12

**Soundness:** 3
**Presentation:** 3
**Significance:** 3
**Originality:** 3
**Overall Recommendation:** 4
**Confidence:** 4

**Summary:**

The paper introduces a Vision-Language-Action (VLA) framework utilizing dual action experts: one for coarse arm transport motions (move) and another for fine-grained, contact-critical behaviors (operate). A learnable router selects the appropriate expert  based on the current task phase (conditioned on contextual features from a shared VLM backbone). To train this router, the authors employ an LLM-based automatic labeling pipeline to generate ground-truth phase selection labels from demonstration videos. Experimental results show that this dual-expert VLA delivers better performance in terms of overall success rate, data efficiency, and training efficiency compared to standard monolithic baselines.

**Compliance With Llm Reviewing Policy:**

Affirmed.

**Key Questions For Authors:**

Questions in “weakness” part above.

5. How to evaluate data quality generated from MLLM?

**Limitations:**

Generalization to unseen tasks may be challenging, since the VLA’s large-scale training data does not include router training, which may limit the model’s generalization ability.

**Strengths And Weaknesses:**

Strengths:

1. Strong Methodological Soundness: The paper presents a comprehensive and well-rounded pipeline. It thoroughly covers the entire process, including the MLLM-based data generation and refinement pipeline, model training, and robust ablation studies that effectively demonstrate both data and training efficiency.


Weaknesses:

1. Limited Architectural Specialization: The proposed methodology is relatively simple, as both the "Move" and "Operate" experts share the exact same network architecture. An opportunity could be to fully leverage their inductive bias; modifying the architectures to be phase-specific (e.g., different capacities or attention mechanisms for transport vs. manipulation) could potentially amplify the advantages of the dual-expert system.

2. Lack of quantitative evalution for router performance: Router is critical for the VLA performance (based on table 3), incorrect prediction of router can lead to significant drop in performance. But quantitative evalution for router performance is missing.
Also in complex bimanual tasks where the arms perform mixed behaviors simultaneously (e.g., one arm is transporting an object while the other is performing fine-grained interaction). It is unclear how the MLLM or the trained router handles these overlapping regimes.

3. Lack of Real-World Validation: The experimental evaluations are conducted entirely on the simulated RoboTwin2 benchmark. The paper lacks evidence on whether the automated labeling pipeline can handle real-world video data and the resulting VLA policy’s performance in real-world tasks.

4. Lack of Validation on Out-of-distribution tasks: It is not clear the VLA performance on unseen objects/tasks, especially whether the router still can correctly predict the labels for dual-experts selection, given that the VLA’s large-scale training data does not include router training.

---

> ### Author Rebuttal · Authors · 2026-03-31
>
> Thank you for your efforts in reviewing this paper. We answered the questions in order and combined similar ones.
>
> A1: For the question of network structure, we appreciate the insightful suggestion regarding heterogeneous experts. Considering the varying action complexities of the two phases, modifying the architectures to be phase-specific is indeed a valuable idea. However, our main focus is data-centric: we investigate whether separating these two phases, which follow different data distributions, can improve overall performance. Furthermore, in our subsequent explorations, we discovered that the model relies on different VLM features across the two phases. The 'Move' phase predominantly utilizes shallow features, whereas deep features contribute only marginally. We are currently exploring structural improvements from a more fundamental mechanistic perspective (where 'move' pertains more to semantics and 'operate' to geometry). Within the scope of this work and existing MoE frameworks, we believe that employing structurally heterogeneous experts is not the primary performance bottleneck.
>
> A2: For the question of quantitative evalution for router performance, We had add experiments. To quantitatively evaluate the router's performance, we employed human experts to annotate the videos of all test results. The experts labeled the ground-truth scene at each inference step, which was then compared against the model's predictions. The final routing accuracy is 92.3%, demonstrating that the router's performance is highly reliable. Furthermore, we observed that the vast majority of failure cases were caused by execution errors (e.g., unstable grasping) rather than incorrect expert selection by the router. Regarding your concern about bimanual tasks with mixed behaviors, we adopted a unified router as a core design principle to maintain generality across both single-arm and bimanual configurations. We indeed observed the 'mixed' scenarios you mentioned—such as in the "place_bread_basket" tasks. Interestingly, our results indicate that during these ambiguous phases (e.g., one arm is operating while the other is transporting), the model spontaneously learned to route such cases to the operate expert. We interpret this as an emergent, safe, and reasonable fallback strategy: the operate expert is capable of handling the increased complexity of mixed states.
>
> A3: For the question of validation of Out-of-distribution tasks, we agree that this is highly valuable; however, for VLA models, transferring to new datasets and sufficiently training the actions is extremely time-consuming. Therefore, to address your concern regarding the router's accuracy, we tested a subset of data from the LIBERO and DROID datasets under a zero-shot setting—that is, using a model trained exclusively on Robotwin. This problem is a challenge for today's methods in both academia and industry. We provided the model with instructions and current observations, recorded whether it predicted a move or operate phase, and compared the predictions against human expert annotations. Ultimately, across 100 randomly sampled instances, the model achieved an accuracy of 68.5% on LIBERO and 67.6% on DROID. LIBERO is a simulator dataset, whereas DROID consists of real-world robot manipulation data, and both are tasks and robots that differ from our training data. We believe these results demonstrate that our model has learned phase-judgment capabilities that can generalize to other tasks and scenarios, proving applicable in both simulated and real-world environments.
>
> A4: For the quetsion of real-world validation, firstly, for automated labeling pipeline, we tested it on Droid using real-world robot videos with the same data as the OOD setup, following the same evaluation protocol as A5. The final tIoU results achieved an accuracy of 78.3%. When predictions within the transition zones are also treated as correct, the accuracy rises to 84.1%. This represents only a marginal decrease, demonstrating the robustness of our labeling approach across different datasets as well as between real and simulated data. For the reasons of lack the policy result in real-world, please refer to wUck A2.
>
> A5: For the question of evaluate data quality generated from MLLM, We utilized human experts to annotate the training data. We evaluated the quality of the phase-specific temporal boundaries generated by the MLLM using temporal Intersection over Union (tIoU), which provides a quantitative measure of temporal localization quality. Moreover, because the boundary between adjacent action phases is often not sharply defined, we additionally annotated transition zones, namely temporal windows that could reasonably belong to either neighboring phase. Under strict evaluation, the MLLM annotations achieve 79.7% accuracy. When predictions within the transition zones are also treated as correct, the accuracy rises to 88.3%, supporting the reliability of the generated training annotations.

---

> > ### Author Rebuttal · Reviewer_AZFp · 2026-04-04
> >
> > Thanks for the rebuttal and the provided technical details. Some concerns have been addressed, while others still remain unresolved.
> >
> > Regarding the response to Q4: accuracies of 68.5% and 67.6% do not indicate strong performance, as a random predictor would already achieve around 50% accuracy. If this gap is attributed to sim-to-real transfer issues, it would be more convincing to include evaluations on different tasks in simulation to better demonstrate the generalization capability of the router.

---

> > > ### Author Response · Authors · 2026-04-07
> > >
> > > Thank you for your continued feedback. We would like to clarify that the previously discussed generalization setting is much harder than a simple binary routing problem. In that setting, the router must handle new robots, new scenes, new objects, and very different visual inputs. It also needs to decide the current stage of the task and select the appropriate control mode. This kind of zero-shot cross-embodiment transfer is still difficult for current VLA systems. We therefore believe the performance drop mainly reflects the difficulty of the setting, rather than a flaw in our routing design.
> > >
> > > To further test the router’s generalization, we evaluated the same model on four new simulation tasks in RoboTwin that were never seen during training. These tasks include unseen objects or novel kinematic patterns. We used the original evaluation protocol and did not apply any retraining or adaptation. The router achieves an **average accuracy of 84.9%** on these tasks. This result shows that the router can still generalize well when the visual domain remains relatively consistent.
> > >
> > > Thank you again for your careful engagement during the rebuttal process. We hope this clarification and the additional results help address your remaining concerns.
> > >
> > > ------------------------------------------------------------------------------------
> > > Additional Details of New Simulation Tasks:
> > >
> > > (Visualizations of these tasks are provided via the following anonymous link: https://anonymous.4open.science/r/MTA_unseen_task_setting-DC18/ )
> > >
> > > place_dual_mugs_pads (Bimanual): Both arms need simultaneously grasp two distinct, randomly positioned mugs and place them onto respective pads.
> > >
> > > place_glue_basket (Single-arm): The arm grasps randomly placed glue bottles of varying types and deposits them into a basket.
> > >
> > > put_tea_cabinet (Bimanual): One arm opens a drawer while the other grasps a tea box and places it inside.
> > >
> > > shake_milk_tea (Single-arm): The arm grasps a randomly placed milk tea cup and executes a shaking motion.

---

### Decision · Program_Chairs · 2026-04-30

**Decision:**

Accept (regular)

**Comment:**

This paper proposes a dual-expert VLA framework that decouples robotic manipulation into a coarse "move" phase and a contact-critical "operate" phase, routed by a learned phase selector trained with MLLM-generated labels. On RoboTwin2 (8 tasks), the method achieves 68.9% average success, a +24% absolute gain over the monolithic pi_0 baseline, with improved data and training efficiency.

The review set is mixed: two reviewers recommend acceptance (scores of 4), while a third reviewer gives a strong reject (score of 2). All three reviewers agree the problem decomposition is well-motivated and the dual-expert design is reasonable. The ablation evidence is a clear strength that random routing drops performance from 69% to 26%, and reversal to 9%, which convincingly shows that learned phase specialization, not added capacity, drives the gains. The shared concerns center on the lack of real-world experiments and limited out-of-distribution evidence.

The rebuttal addressed several issues effectively. The authors provided a manual audit showing 88% label accuracy (with transition tolerance), reported 92% in-distribution router accuracy, and argued persuasively that the extra parameters (0.91% of total) cannot explain a 24% performance jump. However, the cross-domain router accuracy (~68% on LIBERO/DROID, zero-shot) is only modestly above the 50% random baseline, and this measures router classification alone, not downstream task success. No real-robot full-pipeline evaluation was provided. The negative reviewer found these responses insufficient and maintained their score; one positive reviewer was fully resolved, while the other remained only partially resolved on the generalization question.

The core tension here is between a well-executed within-benchmark contribution and a limited evaluation scope. The method works convincingly on its benchmark with strong ablations, but the paper frames itself as a general strategy for human-like manipulation while providing no real-robot results and only thin cross-domain evidence.

**Recommendation: Weak accept**

The algorithm is sound, the ablation evidence is strong, and the practical gains in data and training efficiency are meaningful. The sim-only evaluation on one benchmark is a genuine limitation that prevents a stronger recommendation, but it does not invalidate the methodological contribution. The capacity confound is effectively ruled out, and two confident reviewers support acceptance.